# Small Cajal Body-Specific RNA12 Promotes Carcinogenesis through Modulating Extracellular Matrix Signaling in Bladder Cancer

**DOI:** 10.3390/cancers16030483

**Published:** 2024-01-23

**Authors:** Qinchen Lu, Jiandong Wang, Yuting Tao, Jialing Zhong, Zhao Zhang, Chao Feng, Xi Wang, Tianyu Li, Rongquan He, Qiuyan Wang, Yuanliang Xie

**Affiliations:** 1Department of Urology, Guangxi Medical University Cancer Hospital, Nanning 530021, China; luqinchen@stu.gxmu.edu.cn (Q.L.); 9593@sr.gxmu.edu.cn (J.W.); 2Center for Genomic and Personalized Medicine, Guangxi Key Laboratory for Genomic and Personalized Medicine, Guangxi Medical University, Nanning 530021, China; taoyuting@gxmu.edu.cn (Y.T.); 202120189@sr.gxmu.edu.cn (J.Z.); fengchao@sr.gxmu.edu.cn (C.F.); wangxi@stu.gxmu.edu.cn (X.W.); 3Collaborative Innovation Centre of Regenerative Medicine and Medical Bioresource Development and Application Co-Constructed by the Province and Ministry, Guangxi Medical University, Nanning 530021, China; 4Department of Clinical Laboratory, The First Affiliated Hospital of Guangxi Medical University, Nanning 530021, China; 5Department of Molecular Medicine, Mays Cancer Center, University of Texas Health Science Center at San Antonio, San Antonio, TX 78229, USA; zhangz3@uthscsa.edu.cn; 6Department of Urology, The First Affiliated Hospital of Guangxi Medical University, Nanning 530021, China; litianyu@gxmu.edu.cn; 7Department of Medical Oncology, The First Affiliated Hospital of Guangxi Medical University, Nanning 530021, China; herongquan@gxmu.edu.cn

**Keywords:** small Cajal body-specific RNA12, bladder cancer, ECM, transcription factor, H2AFZ

## Abstract

**Simple Summary:**

Bladder cancer (BLCA) stands as the predominant malignancy within the global urinary system. Small Cajal body-specific RNAs (scaRNAs), constituting a distinct subset of small nucleolar RNAs (snoRNAs), have recently recognized as crucial players in various physiological and pathological processes. The aim of our retrospective study was to elucidate the potential role and mechanism of SCARNA12 in BLCA, offering a foundational understanding the functionalities of scaRNAs. Employing integrated transcriptomic and single-cell proteomic analyses, we comprehensively investigate the impact of SCARNA12 on extracellular matrix (ECM) signaling pathways. Biological experiments provide additional insights into the oncogenic nature of SCARNA12, highlighting its interaction with the transcription factor H2AFZ as a pivotal factor in contributing to the carcinogenesis and progression of BLCA. This study suggests SCARNA12 as a promising diagnostic biomarker and therapeutic target, unveiling its involvement in BLCA through modulation of ECM signaling and interaction with H2AFZ.

**Abstract:**

**Background:** Small Cajal body-specific RNAs (scaRNAs) are a specific subset of small nucleolar RNAs (snoRNAs) that have recently emerged as pivotal contributors in diverse physiological and pathological processes. However, their defined roles in carcinogenesis remain largely elusive. This study aims to explore the potential function and mechanism of SCARNA12 in bladder cancer (BLCA) and to provide a theoretical basis for further investigations into the biological functionalities of scaRNAs. **Materials and Methods:** TCGA, GEO and GTEx data sets were used to analyze the expression of SCARNA12 and its clinicopathological significance in BLCA. Quantitative real-time PCR (qPCR) and in situ hybridization were applied to validate the expression of SCARNA12 in both BLCA cell lines and tissues. RNA sequencing (RNA-seq) combined with bioinformatics analyses were conducted to reveal the changes in gene expression patterns and functional pathways in BLCA patients with different expressions of SCARNA12 and T24 cell lines upon SCARNA12 knockdown. Single-cell mass cytometry (CyTOF) was then used to evaluate the tumor-related cell cluster affected by SCARNA12. Moreover, SCARNA12 was stably knocked down in T24 and UMUC3 cell lines by lentivirus-mediated CRISPR/Cas9 approach. The biological effects of SCARNA12 on the proliferation, clonogenic, migration, invasion, cell apoptosis, cell cycle, and tumor growth were assessed by in vitro MTT, colony formation, wound healing, transwell, flow cytometry assays, and in vivo nude mice xenograft models, respectively. Finally, a chromatin isolation by RNA purification (ChIRP) experiment was further conducted to delineate the potential mechanisms of SCARNA12 in BLCA. **Results:** The expression of SCARNA12 was significantly up-regulated in both BLCA tissues and cell lines. RNA-seq data elucidated that SCARAN12 may play a potential role in cell adhesion and extracellular matrix (ECM) related signaling pathways. CyTOF results further showed that an ECM-related cell cluster with vimentin^+^, CD13^+^, CD44^+^, and CD47^+^ was enriched in BLCA patients with high SCARNA12 expression. Additionally, SCARNA12 knockdown significantly inhibited the proliferation, colony formation, migration, and invasion abilities in T24 and UMUC3 cell lines. SCARNA12 knockdown prompted cell arrest in the G0/G1 and G2/M phase and promoted apoptosis in T24 and UMUC3 cell lines. Furthermore, SCARNA12 knockdown could suppress the in vivo tumor growth in nude mice. A ChIRP experiment further suggested that SCARNA12 may combine transcription factors H2AFZ to modulate the transcription program and then affect BLCA progression. **Conclusions:** Our study is the first to propose aberrant alteration of SCARNA12 and elucidate its potential oncogenic roles in BLCA via the modulation of ECM signaling. The interaction of SCARNA12 with the transcriptional factor H2AFZ emerges as a key contributor to the carcinogenesis and progression of BLCA. These findings suggest SCARNA12 may serve as a diagnostic biomarker and potential therapeutic target for the treatment of BLCA.

## 1. Introduction

Bladder cancer (BLCA) is the most prevalent malignant tumor of the urinary system globally [1,2]. The primary carcinogenic factors associated with BLCA include tobacco smoking and exposure to industrial chemicals [3]. Despite transurethral resection of the bladder tumor (TURBT), followed by chemotherapy or vaccine-based therapy, enabling definitive diagnosis, staging, and primary treatment, the prognosis for BLCA remains unfavorable with a high post-surgery recurrence rate, posing challenges in urologic practice [4]. To benefit more BLCA patients, a comprehensive understanding of the regulatory mechanisms of BLCA and the identification of novel latent prognostic biomarkers are urgently needed.

Recent studies have unveiled that noncoding RNAs (ncRNAs) account for nearly 98% of genome transcripts [5], and these ncRNAs may play pivotal roles in tumorigenesis [6]. Small nucleolar RNAs (snoRNAs), a subtype of ncRNAs ranging in length from 60 to 300 nucleotides and transcribed by RNA Polymerase II (Pol II) [7], have garnered attention. snoRNAs can be classified into three subtypes based on distinct sequence motifs and subcellular locations: small Cajal body-specific RNAs (scaRNAs), box C/D snoRNAs (SNORDs), and box H/ACA snoRNAs (SNORAs) [8,9]. Originally identified for their canonical functions in ribosome biogenesis and RNA modification [10,11,12,13], snoRNAs have been reported to have diverse capabilities, including involvement in chromatin remodeling [14] and other yet-to-be-discovered functions. There is some evidence demonstrating the clinical relevance of snoRNAs, especially in cancer-related events. For instance, SNORD60 played a carcinogenic role in endometrial cancer and regulated PI3K/AKT/mTOR signaling [15]. Additionally, dysregulation of SNORD11B is associated with a poorer prognosis in colorectal cancer patients [16], and knockdown of SNORA42 led to a reduction in tumor proliferation, migration, and invasion in both liver cancer cells and prostate cancer cells [17,18]. Research on the functions and mechanisms of snoRNAs is currently limited, and a more in-depth understanding of the role of snoRNAs in tumors would provide a scientific basis for their theoretical understanding and offer valuable clues for their potential clinical applications.

In this study, we are the first to propose the existence of a novel snoRNA, SCARNA12, which displayed abnormal expression in BLCA. We integrated transcriptomic and single-cell proteomic analyses to comprehensively understand the signaling pathways influenced by SCARNA12. Additionally, we performed a series of biological experiments to uncover the oncogenic roles of SCARNA12 and elucidate its regulatory mechanisms in BLCA. These discoveries could establish a theoretical groundwork for a deeper understanding of the involvement of scaRNAs in carcinogenesis.

## 2. Materials and Methods

### 2.1. Bladder Cancer Samples from Datasets

We employed GEPIA (URL http://gepia.cancer-pku.cn/index.html, accessed on 20 June 2022) to analyze the expression of SCARNA12 across various tumors. Subsequently, we downloaded the BLCA snoRNA expression dataset, comprising 396 tumor tissue samples and 16 normal tissue samples, from the SNORic database (http://bioinfo.life.hust.edu.cn/SNORic, accessed on 20 June 2022). We compared the expression levels of SCARNA12 between the tumor samples and the control group. Similarly, we analyzed SCARNA12 expression in GSE160693&GTEx (52 tumor samples and 9 normal samples) and assessed the difference in SCARNA12 expression between tumor samples and the control group in our cohort (52 tumor samples and 39 control samples). The prognostic value of SCARNA12 in BLCA was evaluated through survival analysis.

### 2.2. Clinical Specimens

A cohort comprising 192 patients, who underwent either transurethral resection of bladder tumor or radical cystectomy and were conclusively diagnosed with primary BLCA by two experienced pathologists, was enrolled in the study based on predefined inclusion criteria. The exclusion criteria ensured that none of the enrolled patients had undergone any chemotherapy or radiotherapy prior to enrollment. These patients were recruited from Guangxi Medical University Cancer Hospital and the First Affiliated Hospital of Guangxi Medical University between 2018 and 2021. Tissue samples from all patients were promptly frozen in liquid nitrogen and then transferred to a −80 °C refrigerator for storage. The duration of follow-up was calculated from the time of diagnosis of bladder cancer to the end of the follow-up.

The current study conducted a retrospective analysis involving 192 patients with BLCA, categorized based on tumor status and SCARNA12 expression. Within this cohort, 52 bladder cancer (BLCA) patients were randomly selected for RNA-seq analysis, encompassing 52 cases of tumor tissues and 39 cases of adjacent tissues. Furthermore, within this subset, 20 tumor tissues underwent random selection for single-cell mass cytometry, and subsequent grouping was stratified based on the median expression value SCARNA12 expression. The clinical details of these 52 BLCA patients are outlined in Appendix A. In parallel, a separate cohort of 140 BLCA patients was included for in situ hybridization assay. This cohort comprised 140 cases of tumoral tissue and 51 cases of adjacent tissue, obtained as paraffin histopathological sections. This study adheres to the internationally recognized Reporting Recommendations for Tumor Marker Prognostic Studies (REMARK) guidelines [19]. Written informed consents were obtained from all participants, and the study approval was obtained from the Ethics Committee of the Guangxi Medical University Cancer Hospital (Approval No. LW2022008).

### 2.3. In Situ Hybridization (ISH) Assay

A custom-built ISH kit was purchased from BOSTER Biological Technology Co., Ltd. (Wuhan, China) and then used for the ISH assay. The sections underwent deparaffinization, rehydration, pepsin treatment, and fixation with 4% paraformaldehyde. Subsequently, hybridization of the samples occurred at 41 °C for 4 h. Blocking of non-specific epitopes was carried out at 37 °C for 30 min. Following blocking, the sections were incubated with biotinylated SCARNA12 probes at 37 °C for 1 h. The sections were then treated with diaminobenzidine hydrochloride (DAB) to visualize immunoreactivity. The staining sections were independently scored by two pathologists who achieved consensus based on the staining intensity and the proportion of positive cells. The scoring system for staining intensity is as follows: no staining, 0 points; weak staining (light yellow), 1 point; medium staining (brown yellow), 2 points; and strong staining (brown), 3 points. The scoring system for positive cells is defined as follows: no positive cells, 0 points; <25% positive cells, 1 point; 26–50% positive cells, 2 points; 51–75% positive cells, 3 points; and >75% positive cells, 4 points. The final ISH score for the staining sections is calculated by summing the scores of staining intensity and positive cells. Detailed information is provided in Appendix A. The probe sequences specific to SCARNA12 are as follows:

5′-AGACTAAGGCGAATGCGACTCCGTGCTCTCTGGCCCTTGG-3′;

5′-CCAGATCAATAGCATTGGTGGCCTTGCCTTCATTTCTGGT-3′;

5′-CCACGGTAGGGCTGGGCACAAGCCACCTGAGCGCAACCTT-3′.

### 2.4. Cell Culture

Human bladder cancer (BLCA) cell lines (T24, UMUC3, SW780, and J82) and the normal bladder epithelial cell line (SV-HUC-1) were procured from the cell bank at the Shanghai Institute of Biochemistry and Cell Biology, Shanghai. All cell lines were cultured in Dulbecco’s Modified Eagle Medium F12 (Gibco, Grand Island, NY, USA), supplemented with 10% fetal bovine serum (Gibco, Grand Island, NY, USA), 100 U/mL penicillin, and 0.1 mg/mL streptomycin (Beyotime, Shanghai, China). Cells were maintained in an incubator at 37 °C with 5% CO_2_.

### 2.5. Construction of SCARNA12 and H2AFZ Knockdown Cell Line Using CRISPR/Cas9 Gene-Editing Technology

SCARNA12-lentiCRISPR v2 and H2AFZ-lentiCRISPR v2 vectors were constructed and lentiviral packaging was performed by Azenta Co., Ltd. (Suzhou, China). SCARNA12 sgRNA1 and sgRNA2 lentiviruses were co-infected into BLCA cells with Polybrene (5 μg/mL, Sigma, New York, NY, USA), according to the manufacturer’s instructions. Stable knockdown of SCARNA12 was then established in T24 and UMUC3 cell lines. H2AFZ sgRNA1, sgRNA2, and sgRNA3 lentiviruses were individually infected into the T24 cell line to generate stable knockdown of H2AFZ. The infected cells were selected with culture media containing puromycin (2 μg/mL). After antibiotics selection, the infected cells were harvested for RNA extraction. The infection efficiency was confirmed by Quantitative Polymerase Chain Reaction. The sgRNA sequences are as follows: SCARNA12 sgRNA1: TGGGGAACTCAGGTGCCCTAG; SCARNA12 sgRNA2: CAAGGGCAGGTCTCAATCCC; H2AFZ sgRNA1: TTCATCGACACCTAAAATCT; H2AFZ sgRNA2: AAATCTAGGACGACCAGTCA; H2AFZ sgRNA3: GATGGCTGCGCTGTACACAG.

### 2.6. RNA Extraction and Quantitative Real-Time Polymerase Chain Reaction (qRT-PCR)

The extraction of total RNA from T24-WT, T24 SCARNA12-KD, T24 H2AFZ-KD, UMUC3-WT, UMUC3 SCARNA12-KD, SV-HUC-1, SW780, and J82 cells was completed using Axygen^®^ AxyPrep Multisource RNA Midiprep Kit (Axygen, Union City, CA, USA) according to the manufacturer’s instructions. Using RT reagent Kit with gDNA Eraser (Takara Bio, Inc., Otsu, Japan), 1 mg RNA of each sample was reverse-transcribed to cDNA. Quantitative PCRs were performed on the Light Cycler 96 PCR system (Roche Diagnostics, Mannheim, Germany) by using SYBR Green (Roche Diagnostics, Mannheim, Germany). The qPCR reactions were carried out using SYBR^®^ Select Master Mix (Roche Diagnostics, Mannheim, Germany), according to the manufacturer’s instructions with the following reaction system: 10 μL qPCR Mix, 5 μL cDNA template, 0.3 μL each primer, and 4.4 μL ultrapure water. The qPCR reaction conditions were: Initial denaturation at 95 °C for 10 min, followed by 40 cycles of 95 °C for 15 s, 60 °C for 30 s, and 72 °C for 30 s. The primers used in the present study are as follows: SCARNA12-F: 5′-CATTTCTGGTGCTGCCCCTA-3′; SCARNA12-R: 5′-AGATCCAAGGTTGCGCTCAG-3′; H2AFZ-F: 5′-GCAGTTTGAATCGCGGTG-3′; H2AFZ-R: 5′-GAGTCCTTTCCAGCCTTACC-3′; GAPDH-F: 5′-GTGAACCATGAGAAGTATGACAAC-3′; GAPDH-R: 5′-CATGAGTCCTTCCACGATACC-3′. All experiments were repeated at least three times. The relative mRNA level was normalized using the GAPDH mRNA level and calculated using the 2^−ΔΔCT^ method with the following formulae: ΔCt = Ct tested gene − average Ct reference gene, ΔΔCt = ΔCt tested well − average ΔCt control well. All procedures for qPCR adhered to the MIQE guidelines [20].

### 2.7. Cell Viability Assay

Cell proliferation was assessed using the MTT assay. T24-WT, T24 SCARNA12-KD, UMUC3-WT, and UMUC3 SCARNA12-KD cells (0.8 × 10^3^) were cultured for 24, 48, 72, and 96 h in 96-well plates. After the respective incubation periods, 10 μL of MTT stock solution was added to each well and incubated for 4 h at 37 °C. Subsequently, 100 μL of dimethyl sulfoxide (DMSO) was added to each well, and absorbance was measured at 450 nm using a microplate reader (BioTek Instruments, Inc., Winooski, VT, USA). The cell viability at different time points was calculated by normalizing the cell viability at defined time points to that at 0 h. The formula used for determining cell relative viability is as follows: Cell relative viability = (tested well OD450 − blank OD450)/(average 0 h well OD450 − blank OD450) × 100%.

### 2.8. Colony Formation Assay

T24-WT, T24 SCARNA12-KD, UMUC3-WT, and UMUC3 SCARNA12-KD cells (800 cells/well) were seeded into 6-well plates and incubated at 37 °C and 5% CO_2_. The medium was changed every 7 days. After two weeks, the cells were washed with PBS, fixed with 100% methanol for 30 min at room temperature, and stained with 0.2% crystal violet for 15 min at room temperature. Following staining, the cells were washed with PBS three times, and colonies were observed under a light microscope and counted.

### 2.9. Wound-Healing Assay

T24-WT, T24 SCARNA12-KD, UMUC3-WT, and UMUC3 SCARNA12-KD cells (7 × 10^4^/well) were seeded in 6-well plates, and wounds were created by making a scratch on the plate using a sterile tip. Following this, cells were washed with PBS and incubated in serum-free culture medium. After the specified duration, the distance between the two wound margins was measured. Data for both WT and SCARNA12-KD cells were recorded at 0, 24, and 48 h, respectively.

### 2.10. Transwell Assay

Transwell experiments were conducted using Millicell Cell Culture Inserts (24-well plates; 8 µm pore size) with T24-WT, T24 SCARNA12-KD, UMUC3-WT, and UMUC3 SCARNA12-KD cells. For the migration assay, 5 × 10^3^ cells in serum-free medium were seeded in the upper chambers. In the invasion assay, the upper chamber membranes were coated with 10 μL Matrigel (Corning 356231) in 80 μL serum-free DMEM-F12 medium for 6 h in a humidified incubator before seeding the cells. The lower chambers contained DMEM-F12 medium with 10% FBS. The cells were incubated for 24 h for the migration assay and 48 h for the invasion assay. Subsequently, cells on the lower membranes were fixed with 4% paraformaldehyde for 15 min at room temperature, stained with crystal violet, and observed at ×100 magnification. Five fields were randomly chosen under a light microscope, and the average number of cells per field was calculated.

### 2.11. Cell Cycle Analysis

T24-WT, T24 SCARNA12-KD, UMUC3-WT, and UMUC3 SCARNA12-KD cells (2 × 10^5^) were plated in 6-well plates and incubated for 48 h. Subsequently, cells were harvested, fixed in 70% ethanol at −20 °C for 24 h, and subjected to cell cycle analysis using the Cell Cycle Analysis Kit (MultiSciences, Hangzhou, China). Flow cytometry was performed on an Accuri C6 Plus flow cytometer (BD Biosciences, San Jose, CA, USA), and ModFit software 5.2 (Verity Software House, Topsham, ME, USA) was employed to calculate the proportion of cells in different phases of this analysis.

### 2.12. Cell Apoptosis Assay

T24-WT, T24 SCARNA12-KD, UMUC3-WT, and UMUC3 SCARNA12-KD cells (2 × 10^5^) were cultured in 6-well plates and harvested after 48 h of incubation. Subsequently, the cells were washed with PBS and resuspended in staining buffer. Following the manufacturer’s protocol, the Annexin V-FITC/PI Apoptosis Kit (MultiSciences, Hangzhou, China) was employed to assess cell apoptosis. Stained cells were examined using the Accuri C6 Plus flow cytometry.

### 2.13. Nude Mice Experiments

BALB/c nude mice (male, 5 weeks old) were obtained from Guangxi Medical University Animal Centre (Nanning, China). All mice were maintained in pathogen-free cages at 26–28 °C. A total of 3 × 10^6^ T24-WT, T24 SCARNA12-KD, UMUC3-WT, and UMUC3 SCARNA12-KD cells were resuspended in 100 μL PBS and injected into the right side of nude mice subcutaneously (5 nude mice per group). For 4 weeks, the survival status and tumor size of nude mice were supervised. After that, all the mice were euthanized and tumors were collected for analysis. Tumor volume was calculated using the following formula: V (mm^3^) = width^2^ (mm^2^) × length (mm)/2. Animal experiments were approved by the Animal Ethics Committee of Guangxi Medical University (No. 202008001).

### 2.14. Chromatin Isolation by RNA Purification (ChIRP) Experiment

Approximately 5 × 10^7^ T24-WT cells were harvested for the ChIRP experiment. Briefly, chromatin-associated RNA was selectively isolated through specific hybridization to biotinylated oligonucleotides targeting SCARNA12. Subsequently, the RNA–protein complexes were crosslinked and purified using streptavidin beads, followed by elution of the biotin–ligated complexes for downstream analysis. After isolation, the RNA–protein complexes can be subjected to silver staining to visualize the protein components. Additionally, the isolated proteins can be further analyzed using liquid chromatography-mass spectrometry (LC−MS) to identify and quantify the proteins present in the complexes.

### 2.15. RNA-Sequencing and Functional Analysis of SCARNA12 in BLCA

Bulk RNA sequencing was performed as previously described [21]. FastQC (URL: https://www.bioinformatics.babraham.ac.uk/projects/fastqc/, accessed on 15 May 2023) was applied to evaluate the quality of the paired-end RNA seq data. Through data pre-processing, Fastp (URL: https://github.com/OpenGene/fastp, accessed on 15 May 2023) can effectively remove adaptor components, correct low-quality bases, and obtain qualified and clean data. Principal component analysis (PCA) was applied to check the repeatability of the experiment. Then, the expression matrix was analyzed with edgeR to obtain differentially expressed genes (DEGs, threshold: absolute value of fold change ≥ 1.5, *p* ≤ 0.05). The Cluster Profiler R package (version: 3.18.0) was used to conduct GO and KEGG enrichment analyses of DEGs. The GO enrichment analysis includes three aspects: biological process (BP), cell component (CC), and molecular functions (MF). The “c2.cp.kegg.v7.5.1.symbols.gmt” gene set from the MSIGDB was downloaded as the reference gene set and gene set variation analysis (GSVA) was performed to evaluate the pathway enrichment of BLCA samples based on gene expression level. The RNA-seq data is published in our previous study [21] and is accessible through the accession code GSE216037 and can be retrieved from the following link: https://www.ncbi.nlm.nih.gov/geo/query/acc.cgi?acc=GSE216037 (accessed on 15 May 2023). 

### 2.16. Transcription Factor Prediction

The BART (binding analysis for regulation of transcription) tool [22] was used to infer the specific transcription factors based on the differentially expressed genes upon SCARNA12 knockdown. The transcription factor prediction was obtained using the average rank of Wilcoxon *p*-value, Z-score, and the maximum AUC among datasets.

### 2.17. Single-Cell Isolation and Metal-Isotope-Labeled-Antibodies

The fresh tumor tissues were minced into small fragments with surgical scissors firstly and dissociated with 1.5 mg/mL collagenase type I (17100017; Gibco, Grand Island, NY, USA) supplemented with 0.2 mg/mL DNase I (10104159001; Roche, Basel, Switzerland) for 40 min at 37 °C. Next, the cell suspension was filtered through a 70 μm cell strainer and lysed in red blood cell (RBC) buffer (Solarbio, Beijing, China) to remove red blood cells. Finally, the single-cell suspension was washed with DPBS three times, resuspended in FBS, complemented with 10% DMSO, and stored at −80 °C.

A panel of 33 antibodies used for CyTOF in this study is listed in Appendix A. Preconjugated antibodies were purchased from the Fluidigm supplier (Fluidigm, San Francisco, CA, USA) directly, while others were conjugated in-house using the Maxpar X8 chelating polymer kit (Fluidigm) according to the manufacturer’s instructions.

### 2.18. Single-Cell Mass Cytometry (CyTOF) and Data Analysis

Twenty BLCA patients who underwent curative resection at Guangxi Medical University Cancer Hospital were included in the study. Initially, four individual samples were barcoded using six stable palladium isotopes (102 Pd, 104 Pd, 105 Pd, 106 Pd, 108 Pd, 110 Pd), as previously described [23]. Each barcoded tube contained 3 × 10^6^ cells and was stained with cisplatin (Fluidigm) to identify live/dead cells. Subsequently, the cells were incubated with metal-conjugated surface-membrane antibodies for 30 min, fixed in 1.6% paraformaldehyde, and permeabilized with 100% methanol (10 min at 4 °C) to permit intracellular staining with metal-conjugated antibodies for an additional 30 min. Finally, the cells were resuspended in an iridium-containing DNA intercalator [24] and incubated for 20 min at room temperature or overnight at 4 °C before analysis on a CyTOF-Ⅱ mass cytometry (Fluidigm). To ensure comparability, signal normalization was carried out using EQ Four Element Calibration Beads (EQ Beads, 201078, Fluidigm) according to the manufacturer’s instructions [25].

Single-cell samples were acquired at approximately 500 events per second on a CyTOF-Ⅱ mass cytometry (Fluidigm). Each sample was normalized to the internal bead standards before analysis. To exclude dead cells and debris, gating was based on Event length and DNA content [26], as well as cisplatin negativity. Data plots, heat maps, and histograms were generated using custom R scripts. For low-dimensional visualization of single-cell data, the t-distributed stochastic neighbor embedding (tSNE) algorithm was employed, and characteristic clusters were identified using phenograph [27]. Analysis was performed on 5000 randomly extracted cells from each sample using the R package.

### 2.19. Statistical Analysis

All experiments were conducted in at least three independent experiments. The SPSS 22.0 software (IBM Corp, Armonk, NY, USA) and GraphPad Prism 8.0 (GraphPad Software, San Diego, CA, USA) were used for statistical analyses. The independent t-test was used to compare the differences between the two groups, analysis of variance (ANOVA) was used to compare three or more groups, repeated-measures ANOVA was used for MTT assay, Spearman rank correlation was used for correlation analysis, and the log-rank test was used for survival analysis. Data were presented as the means ± standard deviation. A *p* value less than 0.05 was considered statistically significant in all analyses (ns: no significant; * *p* < 0.05; ** *p* < 0.01; *** *p* < 0.001).

## 3. Results

### 3.1. SCARNA12 Is Highly Expressed in BLCA Tissues and Cell Lines

Our previous research has highlighted the substantial involvement of the snoRNAs in transcriptional regulation in cancer [28], implying crucial roles in carcinogenesis. Nevertheless, their specific functions remain largely constrained. We initially downloaded the TCGA dataset and assessed the expression levels of SCARNA12 across various types of cancer. As shown in Figure 1A, it is evident that SCARNA12 shows a significant increase in expression in BLCA (bladder urothelial carcinoma) compared to normal tissues. Similarly, it is noticeably upregulated in other invasive cancers such as glioblastoma multiforme (GBM), cholangiocarcinoma (CHOL), and brain lower-grade glioma (LGG) compared to normal tissues.

We further conducted a comprehensive analysis of BLCA datasets from GEO and GTEx to investigate the SCARNA12 expression between BLCA tumors and their normal adjacent tissues. Our analysis revealed a significant upregulation of SCARNA12 in tumor tissues compared to normal tissues, as depicted in Figure 1B,C. Subsequent validation in our bladder cancer cohort confirmed these above findings (Figure 1D). To further verify the reliability of RNA sequencing (RNA-seq), we performed a qPCR experiment to examine the expression level of SCARNA12 in both BLCA tissues and BLCA cell lines. We assessed SCARNA12 expression in four BLCA cell lines (T24, UMUC3, SW780, and J82) and a normal bladder epithelial cell line SV-HUC-1 (Figure 1E). Given the highest expression levels of SCARNA12 observed in T24 and UMUC3 cell lines among the tested cells, they were then selected for further study. In addition, a significant upregulation of SCARNA12 was also observed in BLCA tissues compared to adjacent tissues (Figure 1F). High expression of SCARNA12 in BLCA patients who smoked was shown to be associated with worse survival, indicating that SCARNA12 may serve as a prognostic marker in BLCA (Figure 1G). Additionally, in situ hybridization (ISH) experiments confirmed positive expression of SCARNA12 in BLCA tissues, with localization observed in both the cytoplasm and nucleus (Figure 1H), and the ISH score of SCARNA12 in tumor tissue was notably higher than that in adjacent tissue (Figure 1I, Appendix A). These findings suggest a significant upregulation of SCARNA12 in BLCA, implying its potential contribution to the progression of the BLCA.

### 3.2. SCARNA12 Is Implicated with ECM Signaling and Cell Cycle Regulation

To delve deeper into the biological significance of SCARNA12 in BLCA more comprehensively, we investigated the potential functions mediated by aberrant SCARNA12 expression at the transcriptome level using RNA-seq technology. A total of 52 BLCA patients, collected by our team, were included and stratified into two groups: the high SCARNA12 group and the low SCARNA12 group, based on the expression levels of SCARNA12. The volcano plot depicted 3051 differentially expressed genes (DEGs), including 1224 upregulated and 1827 downregulated genes (Figure 2A). The Gene Set Enrichment Analysis (GSEA) based on the hallmark pathway revealed that high SCARNA12 expression BLCA patients were significantly enriched in signaling pathways such as G2/M checkpoint and E2F targets, which are associated with the cell cycle (Figure 2B). Gene Ontology (GO) function annotation analysis indicated that the differentially upregulated genes were associated with various biological processes, including cell–cell adhesion, cell junction, nuclear matrix, nuclear body, and cell cycle phase transition GO terms (Figure 2C). Consistently, the enrichment results of the KEGG pathway indicate that upregulated differentially expressed genes are significantly enriched in P53, MAPK, apoptosis, and cell cycle, regulating stem cell-related signaling pathways (Figure 2D).

The correlation analysis results further demonstrate a strong positive correlation between the expression of scaRNA12 and well-established genes related to cell cycle and cell adhesion (Figure 2E). It was also observed that cell adhesion-related genes, including PCDHA7, PCDHGB2, and PCDHB10, and cell cycle-related genes such as CDK1, CDC7, and CDC14A, exhibited notably higher expression levels in BLCA patients with high expression of scaRNA12 when compared to those with low expression of scaRNA12 (Figure 2F). The transcriptome-level findings indicate that heightened expression of SCARNA12 in BLCA is linked to signaling pathways associated with extracellular matrix signaling as well as cell cycle signaling.

### 3.3. An ECM-Related Cell Cluster Is Enriched in BLCA with High Expression of SCARNA12 Based on CyTOF Data

To further corroborate the findings derived from RNA-seq analysis, we leveraged single-cell mass cytometry (CyTOF) technology to extend our validation efforts. Our focus was particularly directed toward assessing the correlation between extracellular matrix-related clusters and SCARNA12 expression at the single-cell level. We included 20 BLCA patients from our transcriptome cohort, including 10 BLCA patients with high expression of SCARNA12 and 10 BLCA patients with low expression of SCARNA12, for subsequent CyTOF analysis. A total of 5000 single cells were extracted from each sample for analysis using the t-distributed stochastic neighbor embedding (t-SNE) algorithm, and the t-SNE maps were generated by diverse protein expression patterns among single cells in BLCA samples (Figure 3A). Furthermore, the phenograph algorithm was employed to partition the high-dimensional single-cell data into distinct clusters, each characterized by marker expression patterns (Figure 3B). The analysis of 17 clusters from both the high and low SCARNA12 groups revealed distinct enrichment, suggesting variations in the composition of single-cell subsets, characterizing heterogeneous cellular diversity (Figure 3C,D). We subsequently compared the proportions of these cell clusters, revealing a notable enrichment of cell cluster 2 in the high SCARNA12 group compared to the low SCARNA12 group (Figure 3E,F). Intriguingly, cluster 2 exhibited elevated expression levels of vimentin, CD13, CD44, and CD47, suggesting its characterization as an extracellular matrix (ECM)-related cell cluster. In addition, we discovered that cluster 2 is highly associated with some tumor stemness-related transcription factors: WNT5A, WNT10A, GATA2, and FOSL2 (Figure 3G).

Hence, the higher expression of SCARNA12 in BLCA patients may be associated with increased enrichment of extracellular matrix components, where an ECM-related cell cluster may have a close association with tumor progression.

### 3.4. Knockdown of SCARNA12 Alters Biological Capabilities in BLCA Cell Line

To validate the potential biological roles of SCARNA12 in BLCA, we conducted knockdown experiments targeting SCARNA12 expression in T24 and UMUC3 cell lines. The knockdown efficiency of SCARNA12 is displayed in Figure 4A. MTT assays were performed to assess cell proliferative capacity, which demonstrated a significant inhibition following the knockdown of SCARNA12 in both T24 and UMUC3 cells (Figure 4B). In addition, colony formation assays revealed that knockdown of SCARNA12 resulted in decreased clone numbers in T24 and UMUC3 cells, implying the inhibition of colony formation ability upon SCARNA12 knockdown (Figure 4C,D). Wound healing assays revealed that knockdown of SCARNA12 was associated with reduced wound healing in T24 and UMUC3 cells (Figure 4E,F). In addition, knockdown of SCARNA12 significantly inhibited the motility and invasion ability of T24 and UMUC3 cells in transwell assays (Figure 4G,H). To assess the impact of SCARNA12 knockdown on apoptosis and cell cycle in T24 and UMUC3 cell lines, flow cytometry analyses were performed. In comparison to control cells, bladder cancer cells with SCARNA12 knockdown exhibited a significantly higher percentage of late apoptotic cells, while no significant difference was observed in early apoptotic cells (Figure 5A,C,E,G). These data suggest that the knockdown of SCARNA12 promotes apoptosis in both T24 and UMUC3 cells.

Cell cycle experiments showed a significantly increased percentage of cells were arrested in the G1/G0 and G2/M phases, while a significantly decreased percentage of cells were arrested in the S phase, in SCARNA12-knockdown cells compared with control cells (Figure 5B,D,F,H). These data suggest that knockdown of SCARNA12 leads to cell cycle arrest and the suppression of proliferation in bladder cancer cells. The in vivo effect of SCARNA12 on tumorigenicity was tested by injecting SCARNA12-knockdown and control T24 and UMUC3 cells into four-week-old BALA/c nude mice (five mice per group). After 4 weeks, tumors derived from the SCARNA12-knockdown cells had observably reduced average tumor volume and weight compared with the control group (Figure 5I,J). Taken together, these results suggest that SCARNA12 plays a crucial role in the tumorigenicity, proliferation, migration, invasion, apoptosis, and cell cycle arrest abilities of BLCA cells.

### 3.5. Functional Enrichment Analysis of Target Genes Associated with SCARNA12

The above findings uncovered an essential role of SCARNA12 in the development of BLCA, however, little is known about its impact in regulating signaling pathways and upstream genes. After obtaining the eligible expression profile through the quality control process, all genes were selected for PCA analysis and the results showed that SCARNA12 knockdown had an ideal contribution to grouping samples and ideal biological duplication (Figure 6A). The Gene Ontology (GO) annotation results of 1415 down-regulated DEGs upon SCARNA12 knockdown, including biological processes, cellular components, and molecular functions are shown in a bar plot (Figure 6B). Enriched BP terms mostly include extracellular structure organization, extracellular matrix (ECM) organization, and cell junction assembly. Enriched CC terms mainly contained collagen-containing extracellular matrix, laminin complex, and adheren junction. Enriched MF terms mainly contained integrin binding, receptor-ligand activity, and metallopeptidase activity. The results of the KEGG pathway were mainly included in the cell adhesion molecules, focal adhesion, and ECM–receptor interaction (Figure 6C). Furthermore, GSEA results indicated that cell lines with SCARNA12 knockdown exhibited significant enrichment in pathways related to extracellular matrix degradation and matrix metalloproteinases (Figure 6D). The GO, KEGG, and GSEA pathway analyses revealed significant enrichment in processes relevant to extracellular signaling events. As depicted in the heatmap, the knockdown of SCARNA12 resulted in a pronounced downregulation of expression in ECM-related genes, including CD47, COL5A1, COL6A1, ITGA1, ITGB4, LAMA3, LAMB2, and LAMC1 (Figure 6E). These results imply that SCARNA12 may influence ECM-related signaling in bladder cancer.

### 3.6. Transcription Factor H2AFZ Cooperates with SCARNA12 to Regulate ECM Signaling

To gain further insight into the molecular mechanism that drives SCARNA12 regulatory functions in BLCA, we performed a ChIRP experiment to detect genome-wide SCARNA12-binding proteins with mass spectrometry (ChIRP-MS). Initially, we designed a biotin probe with a reverse complementary sequence to SCARNA12 and conducted a pull-down of biotin–ligated complexes, which enabled the observation of proteins interacting with SCARNA12. As expected, negative control (NC) samples had little protein staining, whereas T24-WT samples presented some clear bands associated with SCARNA12 (Figure 6F). ChIRP-MS for detection of SCARNA12–protein interaction showed that co-factors were mostly nucleic acid binding proteins, enzyme modulators, signaling molecules, and transcription factors (Figure 6G). Top-scoring proteins included H2AFZ, HIST1HID, H3F3C, MYCN, EEF1A1P5, NCL and HNRNPR (Appendix A). These results suggest that SCARNA12 may exert its regulatory functions by interacting with these associated proteins directly.

We applied binding analysis for regulation of transcription (BART) on differentially expressed genes associated with SCARNA12 knockdown to predict transcription factors and found that transcription factor H2AFZ may coordinately regulate the cancer signaling (Figure 6H). Interestingly, our observations revealed a noteworthy enrichment of ECM-related signals among the differentially expressed genes associated with the transcription factor H2AFZ (Figure 6I). This Gene Ontology pattern closely resembles the functions observed upon SCARNA12 knockdown, adding an interesting dimension to our understanding of SCARNA12′s role in modulating ECM-associated signaling. To further investigate the regulatory interplay between H2AFZ and SCARNA12, we also conducted knockdown experiments targeting H2AFZ in the T24 cell line (Figure 6J) and revealed a decrease in the expression level of SCARNA12 following H2AFZ knockdown (Figure 6K). These findings suggest a potential cooperative mechanism between SCARNA12 and H2AFZ, leading to coordinated regulation of target gene expression and subsequent alterations in cellular functions, particularly in ECM processes in BLCA.

## 4. Discussion

Recently, accumulating evidence has shed light on the crucial involvement of small Cajal body-specific RNAs (scaRNAs) in the regulation of various biogenesis processes, including those related to rRNA, tRNA, snRNA, and mRNA [29]. Specifically, scaRNAs are integral components of ribonucleoproteins (RNPs) concentrated in small Cajal bodies, playing a pivotal role in modifying snRNA within these bodies [30]. In addition to their involvement in RNA maturation, there have been reports suggesting that dysregulation of scaRNAs may contribute to the development and progression of various human disorders, including cancers [31,32,33,34,35,36]. Particularly noteworthy is the emerging role of scaRNAs as integral components of exosomes, released by tumor cells to exert key functions within the tumor microenvironment [37]. A recent study illustrated that SCARNA15 loss hampers cancer cell survival, motility, and growth [31]. Moreover, a complicated interaction between SCARNA13 and SNHG10 has also been disclosed in HCC [38]. While existing evidence emphasizes the presence of scaRNAs, limited attention has been given to their biological functions in cancer progression.

Our research marks a pioneering effort, as we have identified a Cajal body-specific RNA, SCARNA12, and conducted a comprehensive study to unveil its role in BLCA. The abnormally heightened expression of SCARNA12 has been discerned in both bladder cancer tissues and cells, as evidenced by data extracted from TCGA, GEO, and GTEx databases. This observation has been further corroborated through qPCR and ISH experiments, thereby establishing a solid scientific foundation for a theoretical comprehension of SCARNA12. RNA-seq analyses were then performed to investigate the potential functions mediated by aberrant SCARNA12 expression. GSEA analysis based on the hallmark pathway revealed that high SCARNA12 expression BLCA patients were significantly enriched in signaling pathways such as G2/M checkpoint and E2F targets, which are associated with the cell cycle [39]. In addition, the upregulated DEGs among BLCA with high expression of SCARNA12 are intricately linked to GO terms such as cell–cell adhesion, cell junction, and nuclear matrix. Additionally, these genes are associated with KEGG pathways, specifically apoptosis, cell cycle regulation, and signaling pathways related to stem cells. It is also observed that a strong positive correlation exists between the expression of scaRNA12 and well-established genes related to cell cycle and cell adhesion, such as PCDHA7, PCDHGB2, PCDHB10, CDK1, CDC7, and CDC14A. Emerging evidence confirms that the abnormal expression of these gene families is involved in carcinogenesis [40,41].

To further substantiate our speculation regarding the role of SCARNA12 in BLCA, we conducted an analysis of CyTOF data and found a significant enrichment of the ECM-related cell cluster 2 in the high SCARNA12 group compared to the low SCARNA12 group. Remarkably, cluster 2 exhibited elevated expression levels of vimentin, CD13, CD44, and CD47, suggesting its characterization as an ECM-related cell cluster [42]. Goo et al. reported the identification of CD13+ cells within the bladder stroma of the lamina propria, forming a discrete cell layer adjacent to the urothelium [43,44]. The transmembrane receptor CD44, known for its affinity to hyaluronic acid, has been implicated in facilitating tumor growth and metastasis, including in the context of bladder cancer [45]. Extensive literature supports the assertion that CD47, identified as an innate immune checkpoint, is markedly expressed in human bladder tumors [46,47,48]. More importantly, cluster 2 exhibits a strong positive correlation with transcription factors WNT5A, WNT10A, GATA2, and FOSL2, all recognized for their involvement in tumor stemness [49,50,51]. These results provide additional support for the proposition that SCARNA12 plays a crucial role in influencing ECM signaling and ECM processes in BLCA.

Biological functional experiments conducted in our study further unveiled the specific effects of SCARNA12 on BLCA cell viability, migration, invasion, cell cycle, and apoptosis in vitro, as well as on tumor growth in vivo, indicating that SCARNA12 plays an oncogenic role in BLCA. To delve deeper into the molecular mechanisms, we employed RNA-seq technology to investigate transcriptional changes following SCARNA12 knockdown. Notably, we observed a significant downregulation of extracellular matrix (ECM)-related signaling pathways upon SCARNA12 knockdown. These results indicate a potential link between SCARNA12 and the regulation of BLCA cell biological behavior through ECM-related signaling pathways. Numerous studies have illustrated the pivotal role of the ECM as a key driver in cancer progression [52,53,54]. The components of the ECM create a cancer-specific microenvironment, triggering biochemical signals that influence cell adhesion and migration [55]. Simultaneously, ECM remodeling serves as a fundamental node for exogenous metabolic regulation through structures such as integrin and collagen [56]. Collagen is renowned for its participation in critical cellular functions such as cell adhesion, migration, tissue scaffold construction, and oncogenesis [57]. This study has confirmed that COL6A1 effectively inhibits the proliferation of MGH-U1 cells, inducing cell cycle arrest in the G1 phase. Additionally, COL6A1 has been shown to inhibit bladder cancer invasion by down-regulating the activities of matrix metalloproteinases 2 (MMP-2) and MMP-9 [58]. Notably, we also noted a down-regulation of COL6A1 in T24 SCARNA12-knockdown cells. Combined with the multi-level evidence chain, we postulate that SCARNA12, as a key molecule, may promote the occurrence and progression of BLCA by regulating ECM-related signaling.

To further investigate the mechanism by which SCARNA12 regulates gene expression in bladder cancer cells, we utilized ChIRP to identify proteins interacting with SCARNA12. Mass spectrometry analysis revealed interactions with histone proteins and transcription factors H2AFZ and MYCN, suggesting a direct interaction of SCARNA12 with chromatin and involvement in transcriptional regulation. Additionally, BART prediction of SCARNA12-associated DEGs highlighted the high-ranking transcription factor H2AFZ. H2AFZ, a highly conserved variant of H2A, is preferentially enriched at the regions of transcriptional start sites, suggesting a relationship between H2A.Z and gene transcription [59]. Recent findings have proposed that H2A.Z acts as a master regulator in the epithelial-to-mesenchymal transition (EMT) process [60]. The regulatory function in EMT is acknowledged to be linked to an epigenetic signature of ECM remodeling [61]. Furthermore, our data on H2AFZ-associated DEGs also indicate their involvement in ECM functions. Therefore, we hypothesize that SCARNA12 may interact with the transcription factor H2AFZ, and this interaction could potentially impact the expression of ECM genes in bladder cancer cells, thus contributing to the development of bladder cancer.

Our findings hold significant implications for future research and clinical applications in BLCA. The identification of the SCARNA12–H2AFZ axis suggests a promising avenue for therapeutic exploration, with the potential to deepen our understanding of specific molecular pathways involved in BLCA progression. Clinically, SCARNA12 could serve as a valuable diagnostic and prognostic marker, as well as an emerging potential therapeutic target for BLCA therapy.

However, our study has certain limitations that warrant consideration. While we innovatively identified an ECM-related cell cluster with specifically high expression of SCARNA12 in BLCA, further investigation is required to understand the potential functions of this specific ECM cell cluster. Our research has primarily focused on human subjects, the cellular level, and simple mouse models. Given the dynamic nature of the tumor microenvironment structure, particularly the extracellular matrix, we recognize the need to establish an in situ mouse model to better understand how SCARNA12 exerts its carcinogenic effects through the extracellular matrix. Moreover, additional experiments are necessary to explore the precise molecular mechanisms underlying the collaborative regulation of bladder cancer development by SCARNA12 and the transcription factor H2AFZ. Validation through external datasets is essential to enhance the robustness and widely applicable understanding of SCARNA12’s roles within the intricate network of bladder cancer development.

## 5. Conclusions

Our study identifies an aberrant alteration of SCARNA12 in BLCA and elucidates its oncogenic roles in regulating biological capabilities within BLCA cells. The higher expression of SCARNA12 in BLCA patients appears to be linked to ECM signaling and an increased enrichment of ECM-related cell clusters. The potential mechanism involves SCARNA12 interacting with the transcription factor H2AFZ, thereby influencing the expression of ECM-related genes and contributing to the development of bladder cancer. These findings expand our understanding of scaRNAs and highlight SCARNA12 as a potential new target for BLCA therapy.

## Figures and Tables

**Figure 1 cancers-16-00483-f001:**
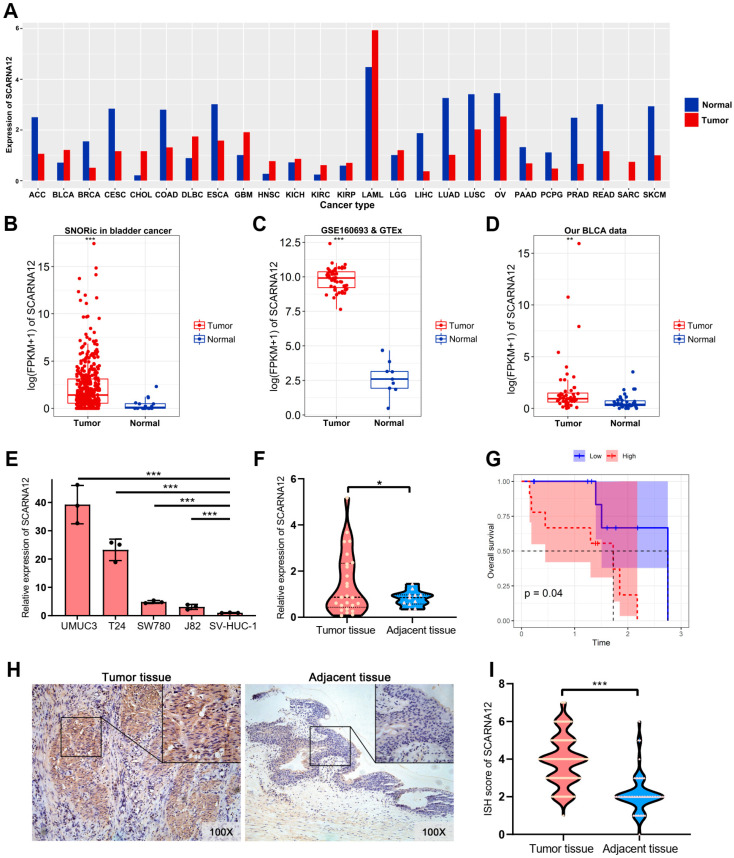
SCARNA12 expression is upregulated in bladder cancer. (**A**) Boxplot shows the expression of SCARNA12 across 23 tumor types using RNA sequencing (RNA-seq) data from GEPIA. The *X*-axis represents tumor types, and the *Y*-axis represents expression value in log2(TPM + 1). *p* values are calculated using *t* tests. (**B**) Boxplots display the expression of SCARNA12 between bladder cancer (BLCA) tissues and normal tissues in SNORic data sets (396 tumor samples and 16 normal samples). (**C**) GSE160693&GTEx data sets (52 tumor samples and 9 normal samples). (**D**) Our own cohort (52 tumor samples and 39 adjacent tissue samples). *p* values are calculated using *t* tests. (**E**) Boxplot displays the expression of SCARNA12 across BLCA cell lines (T24, UMUC3, SW780, and J82) and normal bladder epithelial cell line (SV-HUC-1). *p* values are calculated using the analysis of variance (ANOVA). (**F**) qRT-PCR depicts the expression level of SCARNA12 in 26 tumor tissues and 12 normal tissue samples. *p* values are calculated using *t* tests. (**G**) Survival analysis reveals the effect of aberrant SCARNA12 expression on BLCA patients with smoking history. *p* values are calculated using log-rank tests. (**H**) In situ hybridization (ISH) verifies the different expression levels of SCARNA12 in BLCA tissues and normal adjacent tissue. (**I**) Violin plot shows the ISH score of SCARNA12 in 140 tumor tissues and 51 normal tissues. *p* values are calculated using *t* tests. * *p* < 0.05; ** *p* < 0.01; *** *p* < 0.001.

**Figure 2 cancers-16-00483-f002:**
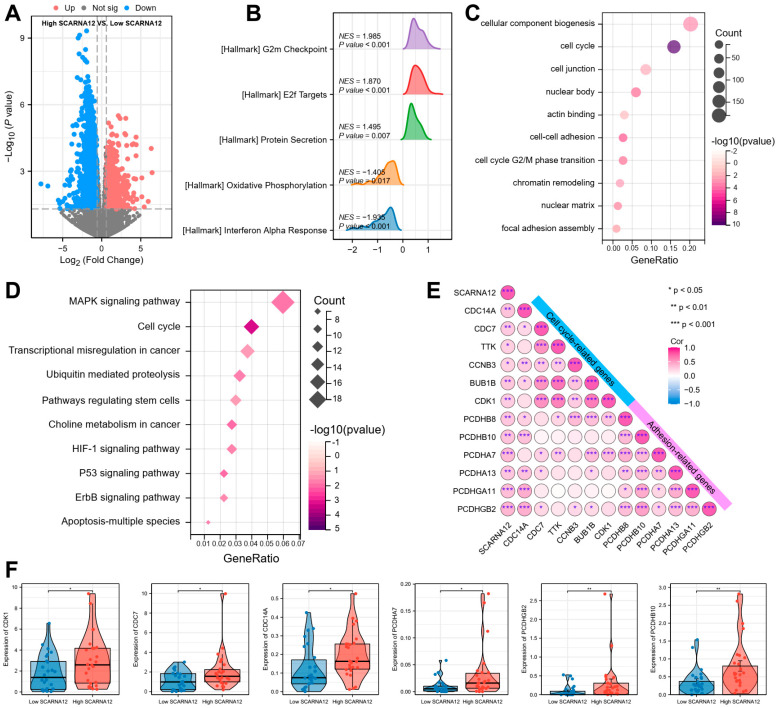
Functional annotations of SCARNA12 in BLCA patients. (**A**) The volcano plot displays differentially expressed genes (DEGs) in BLCA patients between high and low SCARNA12 expression. (**B**) Gene Set Enrichment Analysis (GSEA) plot shows the enrichment of hallmark gene sets among up-regulated DEGs in BLCA with high expression of SCARNA12. (**C**,**D**) Bubble plots visualize representative Gene Ontology (GO) terms and Kyoto Encyclopedia of Genes and Genomes (KEGG) pathways among up-regulated DEGs in BLCA with high expression of SCARNA12. (**E**) The correlation heatmap displays the correlation between SCARNA12 and genes related to cell cycle and cell adhesion. *p* values are calculated using Spearman rank correlation. (**F**) Boxplots show the expression levels of cell cycle- and cell adhesion-related genes in BLCA with high and low expression of SCARNA12. Sample sizes: *n* = 26 for each group. *p* values are calculated using *t* tests. * *p* < 0.05; ** *p* < 0.01; *** *p* < 0.001.

**Figure 3 cancers-16-00483-f003:**
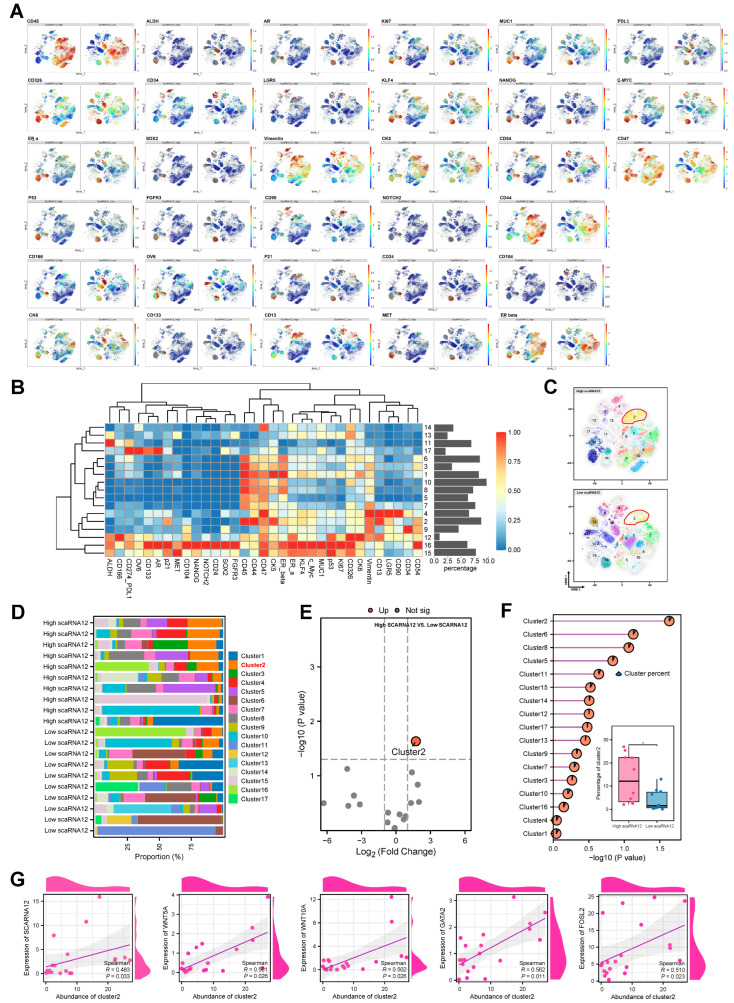
Single-cell Mass Cytometry analysis illustrates tumor microenvironment heterogeneity. (**A**) t-SNE maps display the expression of tumor microenvironment-related markers in BLCA between high and low SCARNA12 expression. Sample sizes: *n* = 10 for each group. (**B**) Heatmap illustrates the expression profiles of tumor microenvironment-related markers across 17 cell clusters. The proportion of cells in each cell cluster relative to the total cell count are shown as bar plots on the right side. (**C**) t-SNE maps display the profiles of cell clusters in BLCA between high and low SCARNA12 expression. (**D**) Histogram indicates the proportions of 17 cell clusters in each BLCA sample. (**E**) Volcano plot depicts the differential abundance of cell clusters in BLCA between high and low SCARNA12 expression. (**F**) Bar plot and boxplot display the proportions of distinct cell clusters and cluster 2 in BLCA between high and low SCARNA12 expression. *p* values are calculated using *t* tests. (**G**) Scatter plot shows the correlation between the abundance of cluster 2 and the expression of stemness-related transcription factors. *p* values are calculated using Spearman rank correlation. * *p* < 0.05.

**Figure 4 cancers-16-00483-f004:**
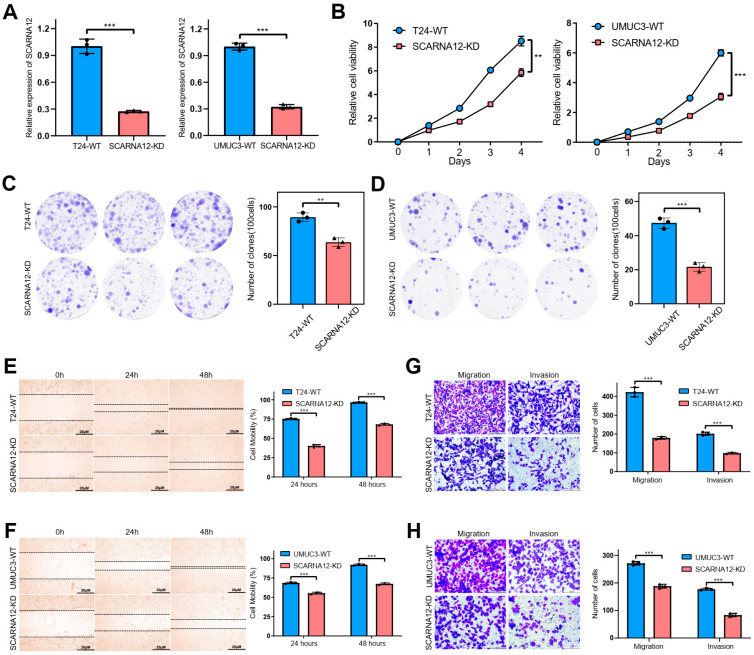
Knockdown of SCARNA12 alters cellular functions in BLCA cell line. (**A**) Knockdown efficiencies of SCARNA12 in T24 and UMUC3 cell lines are assessed by qRT-PCR. *p* values are calculated using *t* tests. Three replicate samples are analyzed for each group, and the same sample sizes are used below. (**B**) Cell proliferation abilities are determined in T24 and UMUC3 WT and SCARNA12-KD cell lines by MTT assays. WT: wild type, KD: knockdown. *p* values are calculated using repeated-measures ANOVA. (**C**,**D**) Clone formation abilities are assessed in T24 and UMUC3 WT and SCARNA12-KD cell lines. Colonies with ≥50 cells and diameter ≥0.5 mm are defined as positive clones. *p* values are calculated using *t* tests. (**E**,**F**) Scratch wound-healing assays reveal the cell-migration abilities in T24 and UMUC3 WT and SCARNA12-KD cell lines. *p* values are calculated using *t* tests. (**G**,**H**) Transwell assays indicate migration and invasion abilities in T24 and UMUC3 WT and SCARNA12-KD cell lines. *p* values are calculated using *t* tests. ** *p* < 0.01; *** *p* < 0.001.

**Figure 5 cancers-16-00483-f005:**
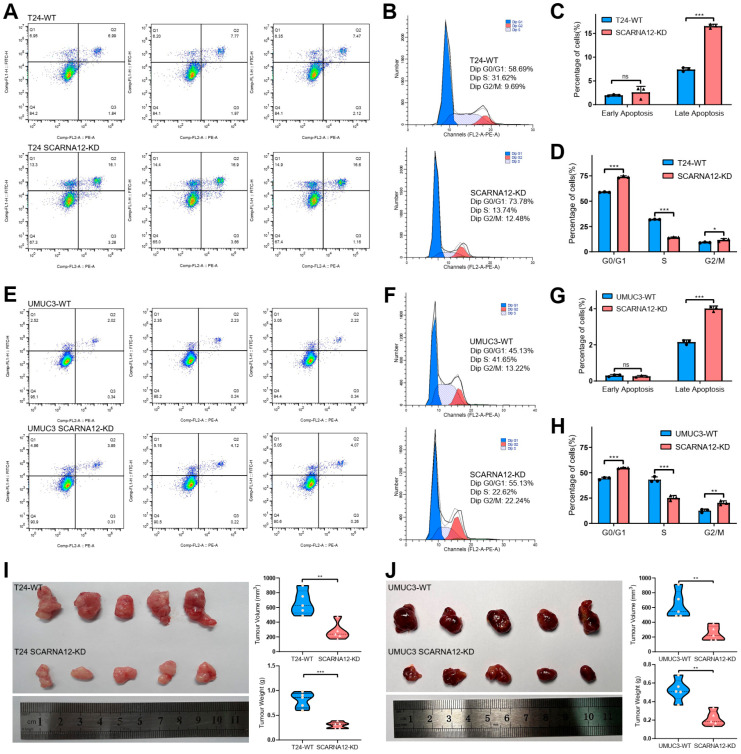
Knockdown of SCARNA12 alters the capabilities of cellular apoptosis, cell cycle arrest, and tumor growth. (**A**,**E**) The effects of SCARNA12 knockdown on cell apoptosis in T24 and UMUC3 cell lines are examined by flow cytometry, and histograms show the percentage of apoptotic cells (**C**,**G**). *p* values are calculated using *t* tests. Three replicate samples are analyzed for each group, and the same sample sizes are used below. (**B**,**F**) The effects of SCARNA12 knockdown on cell cycle arrest in T24 and UMUC3 cell lines are examined by flow cytometry, and histograms show the distribution of cells across different phases of the cell cycle. (**D**,**H**) *p* values are calculated using *t* tests. (**I**,**J**) Photographic images show the xenograft tumors from T24, UMUC3 WT, and SCARNA12-KD groups. Tumor weight and volumes are measured and shown in violin plots. *p* values are calculated using *t* tests. ns: no significant; * *p* < 0.05; ** *p* < 0.01; *** *p* < 0.001.

**Figure 6 cancers-16-00483-f006:**
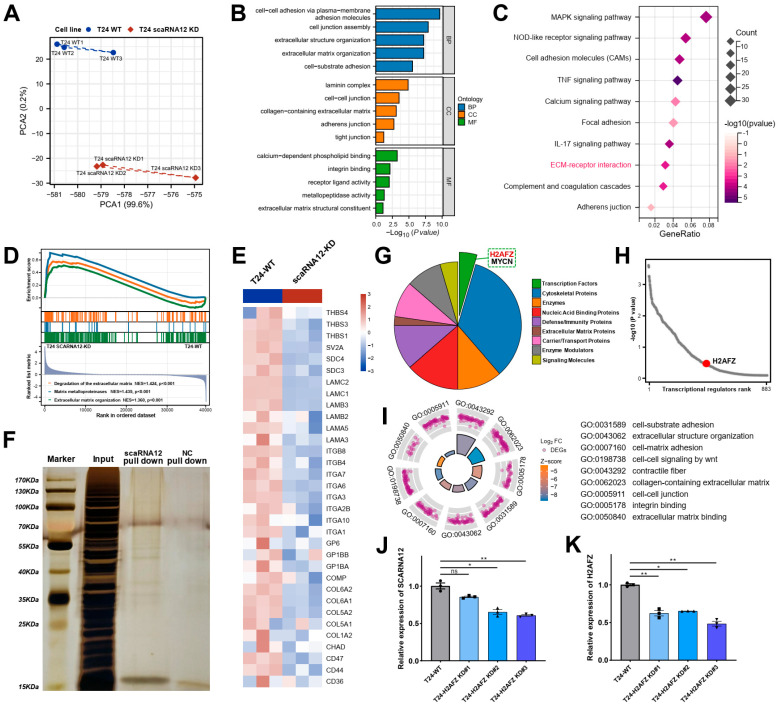
Functional enrichment analysis of target genes and potential interacting proteins associated with SCARNA12. (**A**) PCA plot illustrates the clustering of RNA-seq samples. Three replicate samples are analyzed for each group, and the same sample sizes are used below. (**B**) Bar plot visualizes representative GO terms among down-regulated DEGs in T24 cell lines following SCARNA12 knockdown. The plot categorizes the enriched terms into Biological Process (BP), Cellular Component (CC), and Molecular Function (MF). (**C**) Bubble plots visualize representative KEGG pathways among down-regulated DEGs in T24 cell lines following SCARNA12 knockdown. (**D**) GSEA plot shows the enrichment of extracellular matrix (ECM)-related gene sets among down-regulated DEGs in T24 cell lines following SCARNA12 knockdown. (**E**) Heatmap shows the expression changes of ECM-related genes in T24 cell lines following SCARNA12 knockdown. (**F**) Silver staining shows the proteins potentially interacted with SCARNA12. (**G**) Pie chart shows the components of SCARNA12-interacting proteins. (**H**) Binding analysis for regulation of transcription (BART) for the down-regulated genes in T24 cell lines following SCARNA12 knockdown. (**I**) Circle plot shows the GO terms enriched by H2AFZ-related DEGs. (**J**,**K**) qRT-PCR experiments evaluate the knockdown efficiency of the transcription factor H2AFZ and the expression levels of SCARNA12 after H2AFZ knockdown. *p* values are calculated using ANOVA. ns: no significant; * *p* < 0.05; ** *p* < 0.01.

## Data Availability

The original contributions outlined in the study are publicly accessible, and the data can be accessed at the following link: https://www.ncbi.nlm.nih.gov/ (accessed on 15 May 2023), under the accession code GSE216037. Additional data supporting the conclusions drawn in this manuscript will be provided by the authors upon request.

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
