# Peer review of "Small Cajal Body-Specific RNA12 Promotes Carcinogenesis through Modulating Extracellular Matrix Signaling in Bladder Cancer"

_cancers, 2024, doi:10.3390/cancers16030483_

Round 1

Reviewer 1 Report

Comments and Suggestions for Authors

Very interesting work. My only comment is that the authors should prepare the results/paper in accordance with REporting recommendations for Tumor MARKer prognostic studies (REMARK) criteria and The MIQE guidelines: minimum information for publication of quantitative real-time PCR experiments. This should be supplemented.

Author Response

Dear editors and reviewers,

Thank you so much for your kind efforts into reviewing our manuscript. We highly appreciate and value the insightful comments from both the editors and reviewers, which have greatly improved the quality of the paper. We have read the suggestions carefully and revised the manuscript in accordance with them. Please find the detailed responses and the corresponding revisions/corrections highlighted/in track changes in the attached file.

We wish this paper is suitable for Cancers. Thank you once again for your attention and consideration.

Sincerely,

Yuanliang Xie

Reviewer 2 Report

Comments and Suggestions for Authors

The manuscript describes the oncogenic role of SCARNA12 in bladder cancer.

A broad range of experiments were used in this research. Human cancer databases, cell lines and mouse models show the oncogenic phenotype of SCARNA12. All parts of the manuscript are well written. However, some improvements need to be made in the figures and method description before acceptance of the manuscript. The discussion section could be also improved.

Methods:

2.3 ISH assay – the scoring system is not described. How did you get numbers in the Figure 1I? Criteria for high and low expression are not defined. Which patient samples were used in this method? Data do not correlate with the Clinical Specimens section.

2.5 in the title “H2AFZ” should be added.

2.15 data should be available online and written in the manuscript link to the repository

Results:

Figure 3B What are the percentages on the right presenting?

Figure 4B What is relative cell viability? Error bars are missing.

Figure 5 Channel name should be adjusted

Figures 5B, 5F signal is shifted, x-axis are not the same, which deconvolution method is used?

Figure 6 Figure legend is missing (6H-6K)

General:

-        In all violin/boxplot plots/graphs (1E, 1F, 1I, 3A, 3E, 3F, 3G, 3H, 5C, 5D, 5G, 5H, 5I, 5J, 6J, 6K) samples should be presented with dots (as was presented in Figure 2F).

-        The number of samples should be stated on each figure/graph (n = …)

-        The majority of the figures are really bad quality. It is impossible to read the axis on the graphs.

-        Statistics used in each figure should be stated in the Figure legend

-        The order of the figure number doesn’t follow the text in the results section (e.g. 2c is described in the text after 2E). Some figures need to be rearranged.

-        Font style and size change throughout the manuscript

Comments on the Quality of English Language

The whole manuscript should be carefully read and corrected by many tip-fellers.

Author Response

(The authors gave the same response as above.)

Reviewer 3 Report

Comments and Suggestions for Authors

This study focused on aberrant alteration of SCARNA12 and elucidate its potential oncogenic roles in BLCA via the modulation of ECM signaling. And the interaction of SCARNA12 with the transcriptional factor H2AFZ emerges as a key contributor to the carcinogenesis and progression of BLCA. The article is rich in content, with appropriate experimental methods and reliable results. But there are some issues that need further modification.

1.The latest data should be cited, especially in Introduction first sentence,the cited literature is the 2019 cancer statistics data, and it should be changed. "These ncRNAs may play pivotal roles in tumorigenesis." can be cited from  Front Immunol. 2022 Jan 27:13:803355. 

2.There are still some details in the article that need further repeated proofreading. for example in Introduction last paragraph "These findings may serve as a theoretical foundation for gaining insights into the role of small nucleolar RNAs (scaRNAs) in carcinogenesis." It should be revised.

3. The font size of some text has not been standardized, and these areas must be modified. The parts in Materials and Methods; Results; Discussion.

Author Response

(The authors gave the same response as above.)

Round 2

Reviewer 1 Report

Comments and Suggestions for Authors

The authors did not include detailed information about the guidelins (REMARK) they refer to. Please provide information on where the data regarding the analyzes performed in this work were made available. The data access number GSE216037 indicated by the authors concerns another project. 

  • Tao Y, Li X, Zhang Y, He L et al. TP53-related signature for predicting prognosis and tumor microenvironment characteristics in bladder cancer: A multi-omics study. Front Genet 2022;13:1057302. PMID: 36568387

Please attach files or links to access data regarding calculations using the double delta method. Please provide an example of an anonymized patient's informed consent for genetic testing.

Author Response

Dear Editors and Reviewers,

I hope this letter finds you well. We would like to express our sincere gratitude for your thoughtful review and constructive feedback on our manuscript titled " Small Cajal body-specific RNA12 Promotes Carcinogenesis through Modulating Extracellular Matrix Signaling in Bladder Cancer." Your insightful comments have been invaluable in refining the quality of our work, and we truly appreciate the time and effort you have dedicated to this process.

In response to the feedback received during the initial round of revisions, we have made substantial changes to our manuscript. The first-round modifications have been indicated by red text, and the second-round revisions are highlighted in both red text and yellow background. We have meticulously addressed each comment, aiming to align our manuscript with the standards set by your esteemed journal. Please find the revised files attached in the system.

Additionally, we kindly request replacing current figures with high-resolution PDF versions for improved quality. We believe this adjustment aligns with Cancers’ standards. We hope these revisions meet your expectations. Please let me know if further changes are needed.

Thank you for your continued support and kind consideration.

Sincerely,

Yuanliang Xie

Reviewer 2 Report

Comments and Suggestions for Authors

Thank you for considering all my comments. The manuscript is now much improved and ready for publication.

Author Response

Dear Reviewer,

Thank you for your prompt and constructive feedback. I appreciate your thorough review of the revised manuscript. I am delighted to hear that the changes made in response to your initial comments have resulted in a significantly improved manuscript. Your positive assessment and the acknowledgment that the manuscript is now ready for publication are truly encouraging.

Your expertise has been instrumental in refining the quality of the paper. I am committed to addressing any further suggestions or concerns you may have to ensure the final version meets the highest standards.

Once again, thank you for your time and effort in reviewing my work. I look forward to your continued guidance as we move forward in the publication process.

Best regards,

Yuanliang Xie

Round 3

Reviewer 1 Report

Comments and Suggestions for Authors

The authors made all necessary corrections.